# A Brain Region-Dependent Alteration in the Expression of Vasopressin, Corticotropin-Releasing Factor, and Their Receptors Might Be in the Background of Kisspeptin-13-Induced Hypothalamic-Pituitary-Adrenal Axis Activation and Anxiety in Rats

**DOI:** 10.3390/biomedicines11092446

**Published:** 2023-09-02

**Authors:** Krisztina Csabafi, Katalin Eszter Ibos, Éva Bodnár, Kata Filkor, Júlia Szakács, Zsolt Bagosi

**Affiliations:** Department of Pathophysiology, University of Szeged, P.O. Box 427, H-6701 Szeged, Hungaryfilkor.kata@gmail.com (K.F.);

**Keywords:** kisspeptin, anxiety, corticosterone, hypothalamic-pituitary-adrenal axis, corticotropin-releasing factor, arginine vasopressin, amygdala, hippocampus, stress

## Abstract

Previously, we reported that intracerebroventricularly administered kisspeptin-13 (KP-13) induces anxiety-like behavior and activates the hypothalamic-pituitary-adrenal (HPA) axis in rats. In the present study, we aimed to shed light on the mediation of KP-13′s stress-evoking actions. The relative gene expressions of the corticotropin-releasing factor (*Crf, Crfr1,* and *Crfr2*) and arginine vasopressin (*Avp, Avpr1a,* and *Avpr1b*) systems were measured in the amygdala and hippocampus of male Wistar rats after icv KP-13 treatment. CRF and AVP protein content were also determined. A different set of animals received CRF or V1 receptor antagonist pretreatment before the KP-13 challenge, after which either an open-field test or plasma corticosterone levels measurement was performed. In the amygdala, KP-13 induced an upregulation of *Avp* and *Avpr1b* expression, and a downregulation of *Crf*. In the hippocampus, the mRNA level of *Crf* increased and the level of *Avpr1a* decreased. A significant rise in AVP protein content was also detected in the amygdala. KP-13 also evoked anxiety-like behavior in the open field test, which the V1 receptor blocker antagonized. Both CRF and V1 receptor blockers reduced the KP-13-evoked rise in the plasma corticosterone level. This suggests that KP-13 alters the AVP and CRF signaling and that might be responsible for its effect on the HPA axis and anxiety-like behavior.

## 1. Introduction

Kisspeptins (KPs) are amidated neurohormones part of the Arg-Phe (RF)-amide family with a pivotal role in the central organization of the hypothalamic-pituitary-gonadal axis [1,2]. KP was first isolated from the human placenta as the endogenous ligand of the orphan G-protein coupled receptor GPR54, now referred to as kisspeptin receptor 1 (KISS1R) [3,4]. KP is a 54-amino-acid-long product of the KiSS-1 gene that, via alternative splicing, generates shorter biologically active derivatives containing 14, 13, or 10 amino acids termed KP-14, KP-13, and KP-10, respectively [3,5]. KP and KISS1R are present in a wide range of tissues/organs such as the central nervous system, cardiovascular system, liver, and placenta. Most abundantly, they are expressed in the hypothalamus, more specifically in the arcuate nucleus and the anteroventral paraventricular nucleus (AVPV) in rodents. However, among others, two distinct brain regions also show a moderate-high expression of KP and KISS1R: the amygdala and the hippocampus [6,7,8,9,10], and so far, not much is known about the possible role of KP neurons in these brain regions. Next to KISS1R, KPs can bind to neuropeptide FF receptors (NPFFRs: NPFFR1 and NPFFR2) as well, although with lower affinity than to KISS1R [11,12]. Neuropeptide FF receptor 1 (Npffr1) mRNA is strongly expressed in several brain areas, among them the hypothalamus (e.g., paraventricular nucleus (PVN), periventricular nucleus) as well as in the central amygdaloid nucleus and medial amygdala [13,14,15]. Npffr2-expressing neurons were detected mainly in thalamic and brainstem nuclei, as well as in the hypothalamus [15].

KPs were first investigated in cancer biology as a metastasis suppressor [8], but later multiple studies demonstrated the pivotal role of the KP system in the central regulation of the reproductive axis [2,16]. Furthermore, an increasing body of research suggests that KP signaling might be involved in other neuroendocrine functions, as well as in the modulation of nociception, energy homeostasis, and behavior [17,18,19]. 

Previously we have reported that acute intracerebroventricular (icv) administration of KP-13 evokes an elevation of corticosterone 30 min after treatment and induced anxiety-like behavior in the elevated plus maze test and the traditional open field test, well-known behavioral tests for anxiety [19]. Furthermore, a shorter derivative of KP, KP-8 exerted a similar effect on the hypothalamus-pituitary-adrenal (HPA) axis and anxiety-like behavior [20]. Therefore, the objective of the present study was to further characterize the anxiety-inducing action of KP-13 and investigate its effect on two hormones well-known for the regulation of the neuroendocrine response to stress as well as stress-related behavior. 

Corticotropin-releasing factor (CRF; also referred to as CRH) and arginine vasopressin (AVP) are both crucial regulators of the stress response [21,22,23]. It is well established that stressful stimuli activate the parvocellular neurosecretory cells of the PVN that express CRF as well as AVP, which in turn, via synergistic action on the corticotrophs of the anterior pituitary, cause the release of adrenocorticotropic hormone (ACTH) and, consequently, corticosterone in the adrenal gland of rodents [21,22]. Furthermore, the extra-hypothalamic release of CRF and AVP is involved in coordinating the endocrine and behavioral responses to stress. 

Target cells discern hormonal stimuli by CRF using two distinct receptors: CRF 1 receptor (CRFR1) and CRF 2 receptor (CRFR2), out of which CRF has a high affinity to CRFR1 [22,24]. A plethora of data is available in the literature that attests to the important role of the CRF system in stress responsivity, anxiety, and depression. In fact, *Crfr1* knockout mice show an anxiolytic phenotype [24] that indicates that CRFR1 mediates an anxiety-like action; however, region-specific knockdown of *Crfr1* in globus pallidus externa caused an increase in anxiety-like behavior that highlighted that the effect of CRFR1 activation is brain region-dependent [25]. CRFR2 has been implicated in stress coping and overall CRFR2 activation mediates an anxiolytic effect [26,27]. Nevertheless, it seems that similarly to CRFR1, the effect of CRFR2 activation depends on the brain region. In the medial amygdala, for instance, it mediates anxiety-like behavior [28], whereas in the ventromedial hypothalamus, CRFR2 signaling mediates an anxiolytic effect [29]. 

AVP is also involved in the regulation of stress and associated behavior. AVP can bind to its three distinct receptor subtypes: V1a receptor (V1aR), V1b receptor (V1bR), and V2 receptor [30]. V2 is mainly found in the periphery, more specifically in the renal distal tubules, and is responsible for the antidiuretic action of AVP, whereas V1aR and V1bR next to peripheral expression are also present in the central nervous system and might mediate AVP’s effect on stress, behavior, and mood [21,30,31]. AVP, released in the PVN, has a synergistic effect with CRF on the pituitary ACTH secretion [21]. Furthermore, several literature data point to AVP in the brain mediating an anxiogenic effect. Both the AVP-deficient Brattleboro rats and *V1aR* knockout mice show an anxiolytic phenotype [32,33]. Furthermore, icv administration of AVP induces anxiety-like behavior [34].

CRF and AVP are expressed in abundance in the amygdala and hippocampus. In fact, the highest expression of CRF outside of the hypothalamus is found in the amygdala [35,36]. Furthermore, CRF is coexpressed in subpopulations of hippocampal interneurons throughout the hippocampal layers [37]. These CRF-neurons activate upon stress and mediate stress-induced effects of the hippocampus [37,38]. Hypothalamic AVP-expressing fibers project to the amygdala, which expresses both V1aR and V1bR to exert its stress-inducing effect [39,40]. Also, AVP-producing neurons are found in the amygdala [39]. AVP signaling is also involved in the regulation of hippocampal processes and consequently stress-related behaviors [41,42].

All the above-mentioned data highlight the important role these neuropeptides play in stress response and stress-related behavior. Based on this and our previous experiments [19,20], we hypothesized that KP-13 might alter the CRF and AVP signaling in the amygdala and hippocampus, two brain areas that are involved in the regulation of anxiety-like behavior and express KP and its receptors, as well as CRF and AVP and their receptors [36]. Therefore, the purpose of the present study was to assess if KP influences the CRF and AVP expression in the amygdala and hippocampus and if these two stress hormones might mediate KP’s anxiety- and HPA axis-inducing effects. First, the expressions of *Crf, Crfr1, Crfr2, Avp, Avpr1a,* and *Avpr1b* were measured after KP-13 treatment to assess if KP-13 influences the expression of these genes in the amygdala and hippocampus. Next, we also determined CRF and AVP protein contents in these brain regions. To see if they might mediate KP-13′s anxiety-inducing and HPA-activating effect, animals were pretreated with a non-selective CRF or VP antagonist, after which the behavior of the animals was recorded in a computerized open field test or trunk blood was collected to measure the plasma corticosterone level.

## 2. Materials and Methods

### 2.1. Animals and Housing Conditions

Adult male Wistar rats (Domaszék, Csongrád, Hungary) that weighed 160–250 g were used at the age of 7–8 weeks. They were housed under controlled conditions (12/12 h light/dark cycle, lights on from 6:00 a.m., at constant room temperature) and were allowed free access to commercial food and tap water. The animals were kept and handled during the experiments in accordance with the instructions of the University of Szeged Ethical Committee for the Protection of Animals in Research, which approved these experiments (X./1207/2018). Approximately 180 animals in total were used in our experiments. Every experiment was carried out separately; the same animal was never used for different experimental procedures.

### 2.2. Surgery

The animals were allowed 1 week to acclimatize before surgery. Then, they were implanted with a stainless steel Luer cannula (10 mm long) aimed at the right lateral cerebral ventricle under pentobarbital (35 mg/kg, intraperitoneally) anesthesia. The stereotaxic coordinates were 0.2 mm posterior and 1.7 mm lateral to the bregma, and 3.7 mm deep from the dural surface, according to the atlas of Paxinos et al. [43]. The cannula was secured to the skull with dental cement and acrylate. Further experiments were conducted after a recovery period of 7 days. All experiments were carried out between 8:00 and 10:00 a.m. 

At the end of the experiments, the correct position and the permeability of the cannula were checked. In the behavioral studies, each rat was sacrificed under pentobarbital anesthesia, and in the endocrinological experiments, the head was collected after decapitation. Methylene blue was injected via the implanted cannula and the brains were then dissected. Only data from animals exhibiting the diffusion of methylene blue in all the ventricles were included in the statistical evaluation.

### 2.3. Treatment

Rats were injected with different doses of KP-13 (Bachem Ltd., Bubendorf, Switzerland) dissolved in 0.9% saline icv. in a volume of 2 μL over 30 s with a Hamilton microsyringe, immobilization of the animals being avoided during handling. In the open field test, doses of 0.5, 1, and 2 μg KP-13 were administered, in the case of experiments with antagonists, the most effective dose of KP-13 (1 μg) was applied that was chosen based on our previous experiments [19] and that of the open field test. Antagonist treatment was performed 30 min prior to the peptide challenge. The following antagonists were applied: ⍺-helical CRF(9-41) (Bachem Ltd., Bubendorf, Switzerland), a non-selective CRFR blocker in a dose of 1 μg, and a V1R antagonist (Bachem Ltd., Bubendorf, Switzerland) in a dose of 0.1 μg. The doses of the antagonists were selected based on previous dose–response studies, in which they had no effect per se on the investigated parameters [44,45,46]. Control animals received saline alone. After KP-13 administration, animals were sacrificed at different time points (30 min in case of corticosterone measurement; 2 h in case of gene expression analysis; 4 h in case of protein measurements) or were subjected to behavioral testing. The experimental setup can be viewed in Figure 1.

### 2.4. mRNA Extraction and Quantitative Real-Time PCR

Two hours after icv. KP-13 administration, the animals were sacrificed by decapitation. After isolation of the brain, they were dissected with a pre-cooled adult mouse brain matrix (Ted Pella Inc., Redding, CA, USA). Next, the brains were manually sliced with pre-cooled razor blades in coronal sections (1 mm slots), after which the brain regions were dissected on ice with the guidance of a rat brain atlas [43]. An amount of 1 mm in diameter tissue punches (Ted Pella Inc., Redding, CA, USA) was taken from the amygdala and hippocampus, and placed in Eppendorf tubes filled with 1 mL TRIzol Reagent (Life Technologies, Carlsbad, CA, USA). The tissue samples were immediately frozen and stored at −80 °C until gene expression analysis. To purify and isolate RNA from the samples obtained from the amygdala and hippocampus, samples were homogenized with an ultrasonic homogenizer on ice, and then the total RNA was extracted by using TRIzol extraction protocol and then using GeneJET RNA Purification Kit (Thermo Scientific, Waltham, MA, USA) according to the manufacturer’s instructions. The quality and quantity of extracted RNA were determined by NanoDrop OneC microvolume spectrophotometer (Thermo Fisher Scientific, Waltham, MA, USA). cDNA was synthesized from at least 100 ng of total RNA by using the Maxima First Strand cDNA Synthesis Kit (Thermo Scientific, Waltham, MA, USA) according to the manufacturer’s instructions. SybrGreen technology-based real-time quantitative PCR (CFX96 BioRad) was used to quantify the relative amount of the targeted mRNAs (*Crf, Crf1r, Crf2r, Avp, Avpr1a, Avpr1b*, as well as housekeeping gene *Gapdh*). Specific exon-spanning gene expression assays were used; primer sets are listed in Table 1, respectively. The cycling protocol is listed in Table 2. For controls, we used reaction mixtures without cDNA. Each sample was run in duplicates. The ratio of each mRNA relative to the housekeeping gene was calculated using the 2^−ΔΔCT^ method, and the relative gene expressions were determined for data presentation.

### 2.5. Enzyme-Linked Immunosorbent Assay (ELISA)

CRF and AVP content was measured in brain tissue extracts obtained from the amygdala and hippocampus. Animals were decapitated 4 h after icv. treatment. After isolation of the brain, the amygdala and hippocampus were immediately dissected with a pre-cooled adult mouse brain matrix (Ted Pella Inc., Redding, CA, USA). Next, brains were manually sliced with pre-cooled razor blades in coronal sections (1 mm slots), after which the brain regions were dissected on ice with the guidance of a rat brain atlas [43]. An amount of 1 mm in diameter tissue punches (Ted Pella Inc., Redding, CA, USA) was taken from the amygdala and hippocampus, then placed in Eppendorf tubes and immersed in liquid nitrogen to snap freeze. The samples were stored at −80 °C until the assays were performed. The Pierce Coomassie (Bradford) Protein Assay Kit (Thermo Fisher Scientific, Waltham, MA, USA) was used, according to the manufacturer’s instructions for the measurement of total serum protein concentration. The absorbance was measured at 595 nm with a NanoDrop OneC microvolume spectrophotometer (Thermo Fisher Scientific, Waltham, MA, USA). CRF and AVP were then measured using a competitive CRF and AVP ELISA kit (Phoenix Pharmaceuticals Inc., Burlingame, CA, USA; EK-019-06, EK-065-07), according to the manufacturer’s instruction.

### 2.6. Open Field Test

In the open field test, novelty-induced locomotor activity was assessed via the Conducta 1.0 System (Experimetria Ltd., Budapest, Hungary). The system consists of black plastic open field arenas (inside dimensions: 48 × 48 cm, height: 40 cm) with five horizontal rows of infrared diodes on the walls to register both horizontal and vertical locomotion. The center of each box is illuminated by an LED lightbulb (230 lumens) from above the box. The central zone of the arena is defined as a 24 × 24 cm area in the center of the box (Figure 2). The rats were removed from their home cages and placed at the center of the box 30 min after icv. peptide treatment and their behavior was recorded by the Conducta computer program for 5 min. Six behavioral parameters were measured during the experiment: total ambulation distance, total ambulation time, immobility time, number of rearings (vertical locomotion), time spent in the central zone (central area of 24 × 24 cm), and distance traveled in the central zone (Figure 2). In addition, central ambulation distance/total ambulation distance% and central ambulation time/total ambulation time% were calculated from raw data. The open field experiments were conducted between 8 a.m. and 10 a.m. and the apparatus was cleaned with 96% ethyl-alcohol after each session.

### 2.7. Plasma Corticosterone Measurement

To determine plasma corticosterone concentrations, trunk blood was collected in heparinized tubes. The plasma corticosterone concentration was measured by the fluorescence assay described by Zenker and Bernstein [47] as modified by Purves and Sirett [48].

### 2.8. Statistical Analysis

Data are presented as means ± SEM. The prerequisites of ANOVA were assessed via histograms, skewness, and kurtosis, the Kolmogorov–Smirnov and Levene’s tests. Statistical analysis of the PCR results was performed by Mann–Whitney’s test. For all other data, estimated marginal means were calculated and analyzed by analysis of variance (ANOVA). For the effect of different doses of KP-13 on open field test parameters, one-way ANOVA was employed, followed by the Bonferroni post hoc test for multiple comparisons when the test prerequisites were fulfilled. When the test of the homogeneity of variances was not satisfied, nonparametric ANOVA on ranks (Kruskal–Wallis) was performed, followed by Dunn’s test for multiple comparisons. For the evaluation of ELISA results and all tests with combined treatments two-way ANOVA was performed followed by Bonferroni post hoc test for multiple comparisons. A probability level of less than 0.05 was accepted as indicating a statistically significant difference. The statistical analysis was carried out by SPSS.

## 3. Results

### 3.1. qPCR

The relative expression of *Avp, Avpr1a, Avpr1b, Crf, Crfr1*, and *Crfr2* genes was calculated compared to Gapdh expression and analyzed by the Mann–Whitney test since the normality test and histograms showed a non-normal distribution of the data.

#### 3.1.1. Gene Expression in the Amygdala

In the amygdala, the mRNA expression of *Avp* (Mdn_control_ = 1, Mdn_KP-13_ = 3.646, U = 8, and *p* = 0.0057) and *Avpr1b* (Mdn_control_ = 1, Mdn_KP-13_ = 1.359, U = 20, and *p* = 0.0135) significantly increased, whereas the expression of *Crf* (Mdn_control_ = 1, Mdn_KP-13_ = 0.6726, U = 0, and *p* = 0.0002) was reduced compared to the control group. In the case of *Avpr1a* (Mdn_control_ = 1, Mdn_KP-13_ = 0.9272, U = 36, and *p* = 0.709), *Crfr1* (Mdn_SHAM_ = 1, Mdn_CKD_ = 0.8754, U = 32, and *p* = 0.9999) and *Crfr2* (Mdn_SHAM_ = 1, Mdn_CKD_ = 1.244, U = 36, and *p* = 0.709), no significant difference was detected between the two groups (Figure 3).

#### 3.1.2. Gene Expression in the Hippocampus

In the hippocampus, the relative gene expression of *Crf* (Mdn_control_ = 1, Mdn_KP-13_ = 1.504, U = 6, and *p* = 0.0476) was significantly higher in the KP-13-treated group. On the other hand, *Avpr1a* (Mdn_control_ = 1, Mdn_KP-13_ = 0.7788, U = 0, *p* = 0.0002) mRNA expression showed a marked decrease. In the case of *Avpr1b* (Mdn_control_ = 1, Mdn_KP-13_ = 1.644, U = 5, and *p* = 0.127), *Avp* (Mdn_control_ = 1, Mdn_KP-13_ = 0.7133, U = 12, and *p* = 0.3636), *Crfr1* (Mdn_SHAM_ = 1, Mdn_CKD_ = 0.8939, U = 21, and *p* = 0.69) and *Crfr2* (Mdn_SHAM_ = 1, Mdn_CKD_ = 0.6008, U = 21, and *p* = 0.69), no significant difference was detected (Figure 4).

### 3.2. Enzyme-Linked Immunosorbent Assay

A two-factor analysis of variance on AVP protein level revealed a significant main effect for the treatment factor [F(1,18) = 13.416, *p* = 0.002], region factor [F(1,18) = 22.869, *p* < 0.001]. There was no significant interaction between the two factors [F(1,18) = 1.432, *p* = 0.250), therefore, the effect of the different levels of treatment does not depend on which region is involved. Pairwise comparisons revealed that KP-13 treatment caused a significant increase in the AVP protein level in the amygdala (*p* = 0.002); however, it had no effect in the hippocampus (*p* = 0.125) (Figure 5). 

A two-way ANOVA on CRF protein content showed a significant main effect for the region factor [F(1,17) = 13.235, *p* = 0.003]. However, no significant main effect for the treatment factor [F(1,17) = 0.018, *p* = 0.896]. KP-13 treatment did not affect CRF protein content (Figure 5).

### 3.3. Plasma Corticosterone Level

Two-way ANOVA was conducted to assess the effect of KP-13 treatment and antagonist treatment on corticosterone concentration. Our result showed a statistically significant main effect for the KP-13 treatment [F(1,54) = 19.997; *p* < 0.001] and a statistically significant interaction between the two factors [F(2,54) = 9.058; *p* < 0.001], thus, the effect of KP-13 depends on which antagonist pretreatment was applied. There was no significant main effect for the antagonist treatment factor [F(2,54) = 2.752; *p* = 0.074]. Pairwise comparisons revealed that KP-13 treatment caused a marked elevation in the corticosterone concentration (*p* < 0.001) compared to the saline-treated group. Furthermore, among the KP-13-treated animals, both CRFR antagonists (*p* = 0.025) and V1R antagonists (*p* < 0.001) pretreated were significantly different (Figure 6).

### 3.4. Open Field Test

#### 3.4.1. The Effect of KP-13 on Open-Field Behavior

Univariate ANOVA was used to investigate the effect of KP-13 treatment on the following open-field parameters: total ambulation distance and time, immobility time, rearing activity, central ambulation distance and time, central ambulation distance/total ambulation distance%, and central ambulation time/total ambulation time%. Our result showed that KP-13 has no significant on total ambulation distance [F(3,39) = 0.888; *p* = 0.457] and total ambulation time [F(3,39) = 1.611; *p* = 0.204]. In the case of immobility time [F(3,39) = 2.831; *p* = 0.052], KP-13 showed a tendency to increase; the 2 μg dose of KP-13 significantly increased the immobility time of animals compared to the control (Tukey HSD revealed *p* = 0.048). KP-13 has a significant effect on rearing activity [F(3,39) = 4.368; *p* = 0.01]. Again, the 2 μg dose of KP-13 was the most effective (*p* = 0.007). In the case of central ambulation distance, the test for homogeneity of variance was not satisfied, therefore a non-parametric ANOVA (Kruskal–Wallis) was performed followed by Dunn’s test for multiple comparisons. Results showed that KP-13 treatment significantly decreased the central ambulation distance [Kruskal–Wallis H(3) = 15.831; *p* = 0.001]. Pairwise comparisons with Bonferroni correction revealed that both 1 μg (*p* = 0.001) and 2 μg (*p* = 0.036) of KP-13 significantly decreased the distance traveled in the center of the open field arena. KP-13 had a significant effect on central ambulation time [F(3,39) = 5.6; *p* = 0.003]. Pairwise comparisons revealed that both 1 μg (*p* = 0.006) and 2 μg (*p* = 0.014) of KP-13 significantly decreased the time spent in the center of the open field arena. KP-13 evoked a significant decrease in the central ambulation distance/total ambulation distance% [F(3,39) = 6.367; *p* = 0.001]. Pairwise comparisons showed that both 1 μg (*p* = 0.001) and 2 μg (*p* = 0.023) of KP-13 were significant compared to the control. In the case of the central ambulation time/total ambulation time%, the result was similar [F(3,39) = 5.439; *p* = 0.003]. Again, both the 1 μg dose (*p* = 0.006) as well as the 2 μg dose (*p* = 0.019) of KP-13 was found to be significant (Figure 7 and Figure 8).

#### 3.4.2. The Effect of V1R and CRFR Antagonists on KP-13-Induced Open-Field Behavior

Two-way ANOVAs were conducted to investigate the effect of KP-13 treatment in the presence of CRFR and V1R antagonist pretreatment on open-field parameters. There were no significant changes in the case of ambulation distance [KP-13 treatment: F(1,59) = 0.688; *p* = 0.410, antagonist treatment: F(2,59) = 0.360; *p* = 0.699, interaction: F(2,59) = 0.354; *p* = 0.704] and ambulation time [KP-13 treatment: F(1,59) = 2.725; *p* = 0.105, antagonist treatment: F(2,59) = 0.490; *p* = 0.615, interaction: F(2,59) = 0.598; *p* = 0.553]. In the case of immobility time, there was a significant main effect for the treatment factor (F(1,59) = 5.272; *p* = 0.026), but no significant difference for antagonist treatment or between the two factors [antagonist treatment: F(2,59) = 0.348; *p* = 0.708, interaction: F(2,59) = 0.048; *p* = 0.953]. Pairwise comparisons showed no significant difference between groups. There were no significant changes in the case of rearing activity [KP-13 treatment: F(1,59) = 0.897; *p* = 0.348, antagonist treatment: F(2,59) = 2.660; *p* = 0.079, interaction: F(2,59) = 0.434; *p* = 0.560] (Figure 9). 

However, in the case of central ambulation distance, our result showed a statistically significant main effect for the KP-13 treatment [F(1,59) = 10.665; *p* = 0.002], but no statistically significant main effect for the antagonist treatment factor [F(2,59) = 2.501; *p* = 0.091] or the interaction between the two factors [F(2,59) = 2.724; *p* = 0.075]. Pairwise comparisons revealed that KP-13 treatment caused a marked decrease in the central ambulation distance 30 min after treatment (*p* < 0.001) compared to the saline-treated group. Among the KP-13-treated animals, CRFR antagonist (*p* = 0.08) pretreatment was not significantly different. However, V1R antagonist (*p* = 0.006) pretreatment showed a statistically significant difference (Figure 10).

In the case of central ambulation time, our results showed a statistically significant main effect for the KP-13 treatment [F(1,59) = 15.355; *p* < 0.001] similar to central ambulation distance, but no statistically significant main effect for the antagonist treatment factor [F(2,59) = 2.389; *p* = 0.101] or the interaction between the two factors [F(2,59) = 0.885; *p* = 0.418]. Pairwise comparisons revealed that the KP-13 treatment caused a significant decrease in the central ambulation time (*p* < 0.001) compared to the saline-treated group. Among the KP-13-treated animals V1R antagonist (*p* = 0.021) pretreatment showed a statistically significant difference, however, CRFR antagonist (*p* = 0.071) pretreatment was not significant. In the case of central ambulation distance/total ambulation distance%, again the main effect for KP-13 treatment was found significant [F(1,59) = 6.339; *p* = 0.015], however, no significant main effect was detected for the antagonist treatment [F(2,59) = 1.540; *p* = 0.224] and the interaction between the two factors [F(2,59) = 2.531; *p* = 0.089]. Pairwise comparison showed that KP-13 injection evoked a marked decrease in the central ambulation distance/total ambulation distance% of animals (*p* < 0.001) and among the KP-13 treated groups, the V1R antagonist pretreatment was statistically significant (*p* = 0.011), therefore, the V1R antagonist inhibited the KP-13-induced decrease in central ambulation distance/total ambulation distance%. CRFR antagonist, however, did not alleviate KP-13′s effect (*p* = 0.119). Our results on the central ambulation time/total ambulation time% were quite similar since a statistically significant main effect for KP-13 treatment was detected [F(1,59) = 10.019; *p* = 0.003] and there was no effect for the antagonist treatment [F(2,59) = 1.797; *p* = 0.175] or the interaction [F(2,59) = 1.200; *p* = 0.309]. A pairwise comparison found that KP-13 caused a significant decrease in central ambulation time/total ambulation time% (*p* = 0.003). Furthermore, the V1R antagonist significantly decreased the KP-13-evoked fall in central ambulation time/total ambulation time% (*p* = 0.025), whereas CRFR antagonist treatment among the KP-13-treated animals showed no statistically significant difference (*p* = 0.072) (Figure 10).

## 4. Discussion

Previously, we have reported that KP-13 injection into the lateral ventricle of rats activated the HPA axis and induced anxiety-like behavior in the elevated plus maze and open field tests [19]. In the present study, we aimed to further characterize the anxiety-inducing action of KP-13 and investigate its effect on the expression of two hormones and their receptors, well-known for the regulation of the endocrine, behavioral, and autonomic response to stress: CRF and AVP in the amygdala and hippocampus.

In our experiments, KP-13 caused brain region-specific changes in the gene expression of the AVP and CRF systems. In the amygdala, KP-13 induced a significant upregulation of AVP expression, both at mRNA and protein levels. Although the majority of KP neurons are found in the hypothalamus, a significant population is also present in the amygdala [49]. In rodents, amygdalar KP expression seems to be confined to the medial amygdala (MeA), most prominently to the posterodorsal subnucleus of MeA (MePD). KP neurons in the MeA have been found to maintain reciprocal connections with the accessory olfactory bulb, and project to the hypothalamic GnRH neurons [49]. So far, studies have focused on investigating the role of these KP neurons in the regulation of the reproductive axis. Literature data suggest that the stimulation of KP neurons in the MePD increases the LH pulse frequency possibly mediated by both GABAergic and glutamatergic signaling [50], and sexual behavior [17]. Nevertheless, the MeA also plays a role in the processing of emotional signals, therefore, it might be involved in the mediation of anxiety [51]. In the MeA, a sexually dimorphic population of AVP neurons has been detected, which sends direct and indirect projections to the hypothalamic PVN and triggers greater recruitment of AVP neurons in the PVN following stressful stimuli [40], consequently leading to increased stress responsiveness. Since vasopressin fibers have been found in close apposition with KP neurons in the MePD [49], KP might play a direct role in the activation of AVP neurons in the amygdala. In addition, AVP-expressing neurons in the amygdala are under the control of circulating gonadal steroids, especially in male rodents [52]. In fact, in castrated male mice, treatment with exogenous testosterone has induced hypomethylation of the AVP promoter in the MePD and the bed nucleus of the stria terminalis (BNST), thereby increasing AVP expression [52]. Furthermore, AVP neurons in the MePD have also been involved in integrating pheromonal and hormonal information and regulating sexual behavior. In male Wistar rats, AVP neurons in the MePD were activated following exposure to an inaccessible female [53]. Since the activation of KP neurons leads to the activation of the reproductive axis and consequently the elevation in circulating gonadal steroids [2,16], it is also possible that KP-13′s action on the amygdalar AVP expression is mediated indirectly by testosterone. Furthermore, circulating gonadal steroids, in turn, positively modulate KP expression in the amygdala [54,55]. 

Kp-13 also induced upregulation of *Avpr1b* in the amygdala. V1bRs have been implicated in anxiety since treatment with a selective nonpeptide V1b antagonist (SSR149415) has exerted an anxiolytic-like and antidepressive-like effect in a battery of behavioral tests in mice [56]. Likewise, microinfusion of SSR149415 into the basolateral amygdala has reduced anxiety-like behavior in male rats [57]. Overall, an increased *Avp* and *Avpr1b* expression is in accordance with our previous result that KP-13 induces anxiety-like behavior in rats [19,20].

Interestingly, the expression of amygdalar *Crf* decreased in response to KP-13. According to the literature, amygdalar CRF has mostly been associated with anxiety. For instance, the activation of CRF-expressing neurons in the central amygdala using cre-dependent AAV-DREADD in CRF-cre mice has induced anxiety-like behavior, whereas the inhibition of these neurons has reduced anxiety [36]. It should be noted that although recently a novel population of GABAergic CRF neurons has: been described in the lateral central amygdala (CEA) which sends projections to the ventral tegmental area (VTA) and exerts an anxiolytic-like effect, likely mediated by CRFR1 receptors [58,59]. Since whole amygdala samples were used in our gene expression study, it cannot be determined which amygdalar neuron population was *Crf* downregulated. Nevertheless, it must be noted that whereas *Crf* gene expression was downregulated in the case of KP-13 icv injection, no significant change was detected in the protein level of CRF in the amygdala. 

In the hippocampus, KP and KISS1R are expressed in high densities in the granule cell layer of the dentate gyrus and have also been detected in a low density in the pyramidal cells of CA1 and CA3 [38,60]. It was thus suggested that KP signaling might be involved in hippocampal functions such as learning and memory. Central administration of KP-13 facilitate passive avoidance consolidation [61] and it seems to alleviate amyloid-beta neurotoxicity in the hippocampus [62,63]. Additionally, KP has been implicated in the regulation of BDNF expression and neurogenesis in the hippocampus [64,65], but its connection to CRF, AVP, and their receptors has not been explored in this region yet. In our study, a significant upregulation of *Crf* and downregulation of *Avpr1a* was found following KP-13 treatment in the hippocampus. CRF is expressed in basket-type GABAergic interneurons of the pyramidal cell layer [38,60] and the upregulation of *Crf* in the hippocampus has been linked to the development of anxiety-like behavior. In juvenile rats, moderate psychological stress has evoked the activation of pyramidal cells, which could be significantly dampened by the administration of a CRFR1 antagonist [38]. In another study involving young rats, restraint stress on post-natal day 18 has induced hippocampal activation, demonstrated by upregulation of Fos in CA3 and pCREB in CA1, CA3, and the dentate gyrus, which could be prevented by treatment with a CRFR1 antagonist [66]. Furthermore, the upregulation of CRF and CRFR1 in the CA1 and CA3 regions has also been observed in rats displaying extreme behavioral response (i.e., strong anxiety-like behavior) 7 days after exposure to predator scent [67]. 

We also demonstrated a downregulation of *Avpr1a* in the hippocampus. V1aRs have been detected on the GABAergic interneurons of the dentate gyrus, CA3, CA2, and CA1 regions [68,69]. In the pyramidal cells of the CA1 region, a dose-dependent increase in the frequency of IPSCs has been found in response to AVP, mediated by V1aR activation [70]. There are several studies that assign an anxiety-inducing effect for V1aR signaling. Bielsky et al. found a reduction in anxiety-like behavior in V1aR KO mice [71]. V1aR antagonist treatment also resulted in an anxiolytic effect [72], but the role of hippocampal V1aRs has not been investigated in anxiety yet. 

Overall, our gene expression results suggest that KP-13 does, indeed, affect the expression of *Crf, Avp*, and their receptors in a brain region-dependent manner since, for instance, *Crf* expression in the amygdala decreased, whereas in the hippocampus, it increased. Similarly, *Avp* expression showed a definite elevation in the amygdala and no change in the hippocampus. 

To further establish the connection between CRF and AVP signaling pathways and KP-13′s anxiety- and HPA axis-stimulating effect, we conducted a set of experiments in which, before the administration of KP-13, a pretreatment with non-selective CRFR or a V1R antagonist was performed. 

First, we performed a computerized open field test to underlie KP-13′s anxiety-inducing effect. Different doses of KP-13 were injected icv into male Wistar rats, the behavior of which were then recorded in a non-familiar environment in an open field box. Our results showed that KP-13 dose-dependently reduced the distance traveled and time spent in the center of the arena, which corresponds to anxiety-like behavior. The dose–response curve showed a U-shape that is often seen in the case of peptides [73,74]. The possible mechanism could be homologous desensitization by G-protein coupled receptor kinases that phosphorylate already activated receptors thus lowering the responsiveness of the cell specifically to ligands of those receptors or receptor downregulation [73,74]. Furthermore, higher doses of KP-13 could bind to and activate less specific receptors that might oppose the KISS1R- and/or NPFF receptor-mediated response [73,74]. KP-13 also caused increased immobility time and decreased rearing activity. These underlie the anxiety-inducing effect of KP-13 since they suggest increased freezing and decreased exploratory behavior that is characteristic of anxiety in rodents [75]. 

After we demonstrated KP-13′s effect in this computerized open field test, we pretreated the animals with either ⍺-helical CRF(9-41) or V1R blocker. Our results showed that the V1R antagonist reduced the KP-13-evoked decline in central ambulation distance and central ambulation time, whereas the non-selective CRFR antagonist had no effect. This indicates that AVP signaling pathways might mediate KP-13′s anxiety-inducing effect. Taking these results together with that of the qPCR and the ELISA, the upregulation of AVP signaling in the amygdala and the downregulation of V1aR in the hippocampus might be involved in the anxiety-like behavior induced by the icv administration of KP-13. It must be mentioned that KP’s effect on anxiety-like behavior is somewhat contradictory in the literature. In fact, some studies found KP to exert an anxiolytic effect [76,77], some reported no effect [78,79], and some indicated an anxiety-like effect [19,20,80]. The reason for these discrepancies might lie in the differences in the experimental setup. For instance, in the experiments of Rao et al. and Comninos et al., KP injection was peripheral [78,79], therefore, it is possible that the KP concentration in stress-related brain areas did not reach sufficient levels to induce stress-related behaviors. In addition, it is not so surprising to see different effects in the case of systemic administration and local selective activation of neurons [77]. Also, a confounding factor is that KP’s effect on stress-related behavior was investigated in different species. Ogawa et al. showed an anxiolytic effect in zebrafish, whereas studies that ascribed an anxiogenic effect to KP were performed in rodents [12]. An additional explanation for the contradictory results in KP’s effect on stress-related behavior could be that different forms of KP analogs were administered, and since the affinity of KP to the NPFF receptors is determined by the length of the peptide [12], it is also plausible that the different KP analogs might exert different actions. Nevertheless, our present results are in accordance with our previous results [19,20] as well as the findings of Delmas et al. [80], who reported an anxiolytic phenotype in Kiss1r KO animals.

In our previous study, we demonstrated that corticosterone levels of rats elevated 30 min after icv KP-13 treatment [19]. In our present study, we wanted to determine if CRF and/or AVP signaling might be mediating KP-13′s effect. It is well established in the literature that both CRF and AVP are crucial for the activation of the HPA axis and consequently for the corticosterone response. In fact, CRF and AVP are released from the PVN and they synergistically induce the release of ACTH in the pituitary [21]. Since, in our experiments, KP-13 altered the expression of both AVP and CRF, it is plausible that it also activates the HPA axis directly or indirectly via these two hormones. Therefore, a set of animals was pretreated with either ⍺-helical CRF(9-41) or V1R antagonist 30 min before the KP-13 challenge. Our results showed that KP-13 induced a robust increase in the plasma corticosterone level, the effect of which was inhibited by both antagonists suggesting that both CRF and AVP are involved in mediating KP-13′s effect on the HPA axis. This is in harmony with the literature data mentioned before that both CRF and AVP are involved in the regulation of the HPA axis [21]. 

It is somewhat difficult to assess the clinical significance of our findings in rodents. Our results taken together with those of others indicate that kisspeptins have an influence on the regulation of the HPA axis and stress-associated behaviors. Thus, it is also possible that changes in KP signaling might be involved in the development of HPA axis-related pathologies (e.g., major depression, post-traumatic stress disorder, and anxiety disorders) [81,82]. Several clinical trials are already on the way that are based on the modulation of KP signaling, therefore it is crucial to establish the full range of KP’s biological action and must be taken into account that it might influence mood [83]. Nevertheless, further extensive research is needed to fully establish the mechanism of KP’s action on the HPA axis and stress-related behaviors.

Our results must be interpreted in the context of several limitations of our study. First, in our study, only male Wistar rats were used to avoid the sex differences and the effect of the hormonal changes associated with the estrous cycle. Notably, HPA axis responsiveness shows marked sex differences [84]. In fact, in rodents, females exhibit a more notable corticosterone response in the presence of any stressors, owing to the circulating estradiol levels [84]. Therefore, animal models of both sexes would have yielded more translatable results, and in the future, KP’s effect on the HPA axis and stress-related behaviors in females must be addressed as well. 

Furthermore, in the present study, we have investigated the effect of KP-13 on the gene expression of AVP and its receptors as well as CRF and its receptors. However, only AVP and CRF were investigated at the protein levels. 

Protein levels vary on a high dynamic range, regulated by the rate of translation and degradation, the former being the determining factor [85]. The rate of translation is regulated by multiple mechanisms, only one of which is transcription. Other mechanisms include the activity of eukaryotic initiating factors (EIFs), structural features (e.g., internal ribosome-entry sequences, upstream open reading frames, and secondary or tertiary RNA structures), RNA-binding proteins, as well as microRNAs and small interfering RNAs that are involved in the regulation of translation [86]. Any of these mechanisms might be in play, and thus, our gene expression results on the receptors might not translate to the protein levels. 

Also, it must be noted that other brain regions may also be involved in mediating KP-13′s effect on the HPA axis and anxiety-like behavior. The amygdala and hippocampus were chosen due to the distribution data of KP and KISS1R and the crucial role these regions play in the regulation of the neuroendocrine stress system [10,24]. KP and KISS1R are expressed in other brain regions as well, however, at a much lower expression level [6,10]. Still, it is possible that KP exerts its effect via another route.

## 5. Conclusions

In conclusion, KP-13 seems to alter the expression of *Avp, Crf,* and their receptors in a region-dependent manner. In the amygdala, the KP-13 treatment upregulated the expression of Avp and *Avpr1b* and downregulated the expression of *Crf*, whereas in the hippocampus, KP-13 caused the mRNA level of *Crf* to increase and the mRNA level of *Avpr1a* to decrease. A significant rise in AVP protein content was also detected in the amygdala. Furthermore, KP-13 evoked an anxiety-like behavior in the open field test, that was antagonized by the V1R blocker. In the case of the HPA axis, both CRFR and V1R antagonists reduced the KP-13-evoked rise in the plasma corticosterone level. All these data suggest that KP-13 could affect the AVP and CRF signaling pathways and that might be responsible for its effect on the HPA axis and anxiety-like behavior.

## Figures and Tables

**Figure 1 biomedicines-11-02446-f001:**
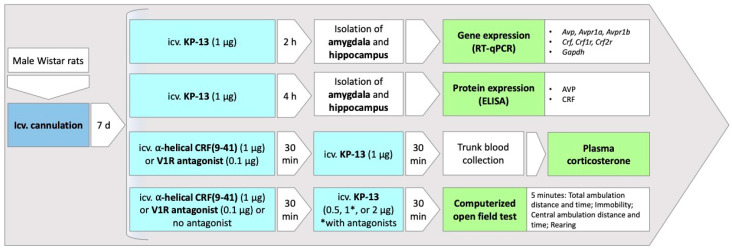
Experimental setup. Abbreviations: icv.: intracerebroventricular; KP-13: kisspeptin-13; V1R: V1 receptor; Avp: Arginine vasopressin, Avpr1a: arginine vasopressin receptor 1A, Avpr1b: arginine vasopressin 1B, Crf: corticotropin-releasing factor, Crfr1: corticotropin-releasing factor receptor 1, Crfr2: corticotropin-releasing factor receptor 2, Gapdh: glyceraldehyde 3-phosphate dehydrogenase.

**Figure 2 biomedicines-11-02446-f002:**
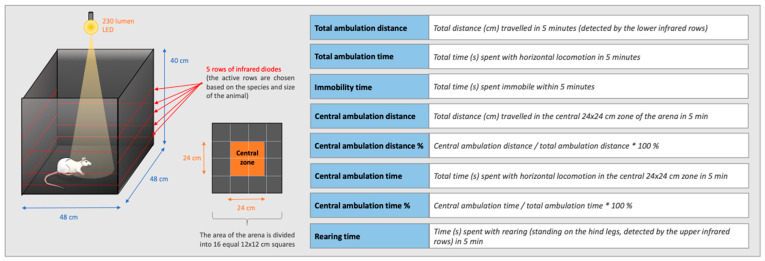
Schematic presentation of the open field box and the open-field parameters.

**Figure 3 biomedicines-11-02446-f003:**
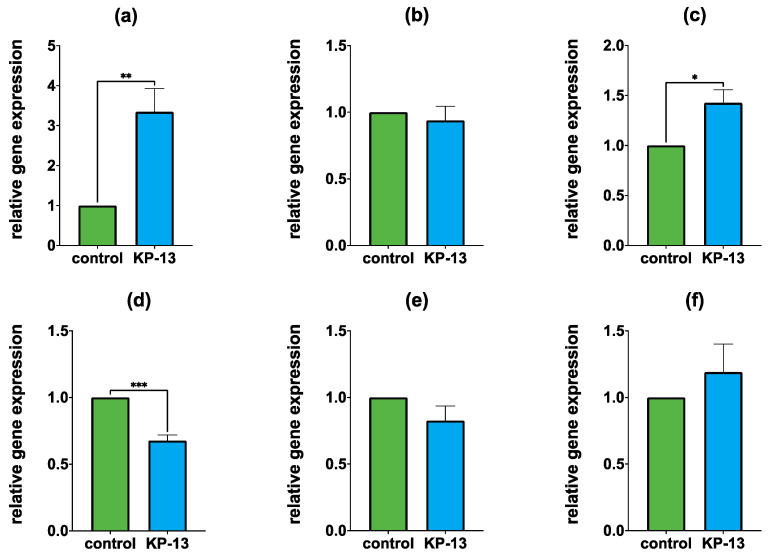
Relative gene expression in the amygdala: (**a**) *Avp*; (**b**) *Avpr1a;* (**c**) *Avpr1b;* (**d**) *Crf;* (**e**) *Crfr1;* and (**f**) *Crfr2*; mean + SEM, *n* = 8–9; * *p* < 0.05, ** *p* < 0.01; *** *p* < 0.001; Abbreviations: Avp: arginine vasopressin, Avpr1a: arginine vasopressin receptor 1A, Avpr1b: arginine vasopressin receptor 1B, Crf: corticotropin-releasing factor, Crfr1: corticotropin-releasing factor receptor 1, Crfr2: corticotropin-releasing factor receptor 2, KP-13: kisspeptin-13.

**Figure 4 biomedicines-11-02446-f004:**
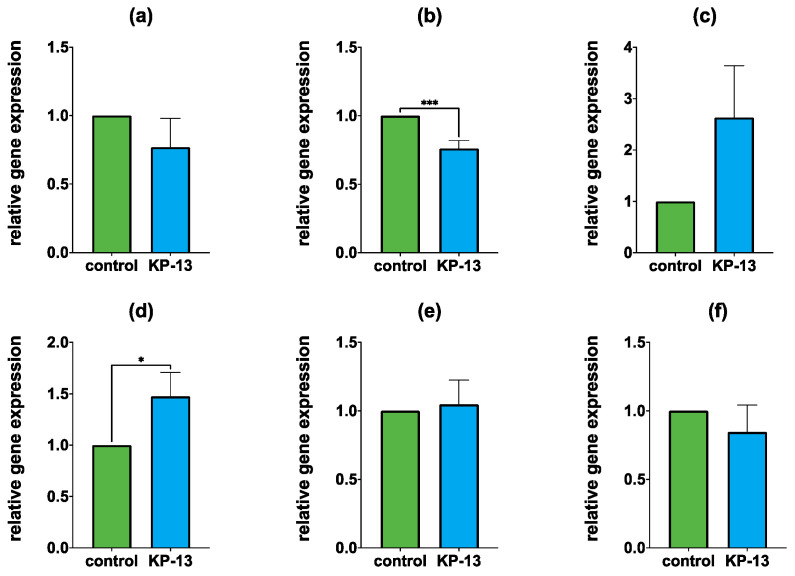
Relative gene expression in the hippocampus: (**a**) *Avp*; (**b**) *Avpr1a*; (**c**) *Avpr1b*; (**d**) *Crf*; (**e**) *Crfr1*; (**f**) *Crfr2*; mean + SEM, *n* = 5–8; * *p* < 0.05, *** *p* < 0.001; Abbreviations: Avp: arginine vasopressin, Avpr1a: arginine vasopressin receptor 1A, Avpr1b: arginine vasopressin receptor 1B, Crf: corticotropin-releasing factor, Crfr1: corticotropin-releasing factor receptor 1, Crfr2: corticotropin-releasing factor receptor 2, KP-13: kisspeptin-13.

**Figure 5 biomedicines-11-02446-f005:**
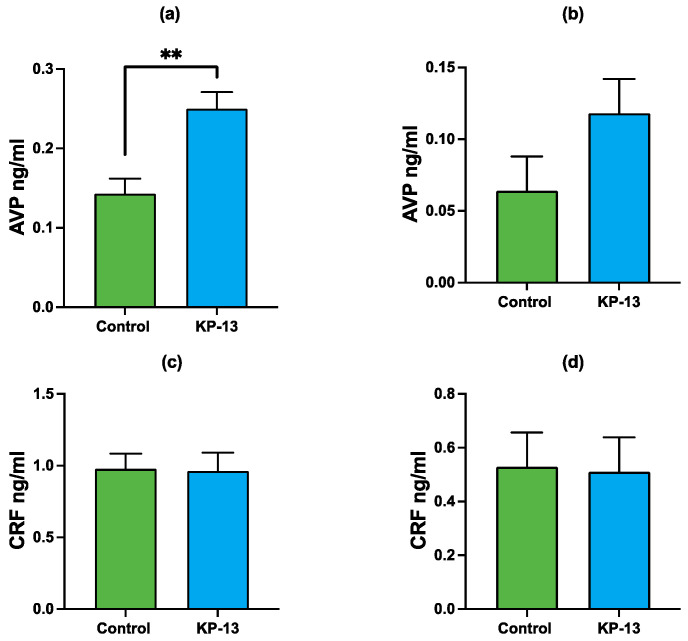
Arginine vasopressin and corticotropin-releasing factor protein expression in the amygdala and hippocampus: (**a**) AVP in the amygdala; (**b**) AVP in the hippocampus; (**c**) CRF in the amygdala; (**d**) CRF in the hippocampus; mean + SEM, *n* = 4–6; ** *p* < 0.01; Abbreviations: AVP: arginine vasopressin, CRF: corticotropin-releasing factor, KP-13: kisspeptin-13.

**Figure 6 biomedicines-11-02446-f006:**
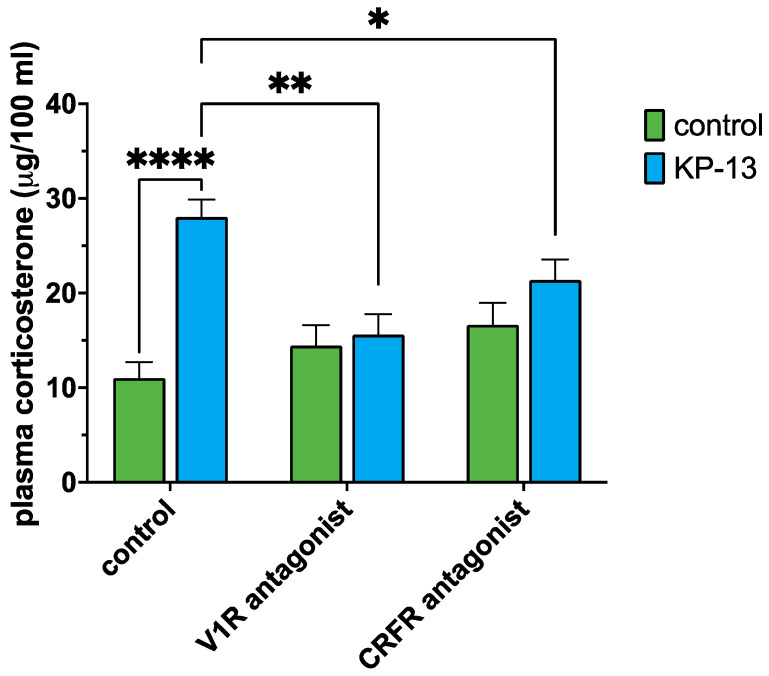
Plasma corticosterone results: mean + SEM, *n* = 7–13; * *p* < 0.05, ** *p* < 0.01, **** *p* < 0.0001; Abbreviations: V1R: V1 receptor, CRFR: corticotropin-releasing factor receptor, KP-13: kisspeptin-13.

**Figure 7 biomedicines-11-02446-f007:**
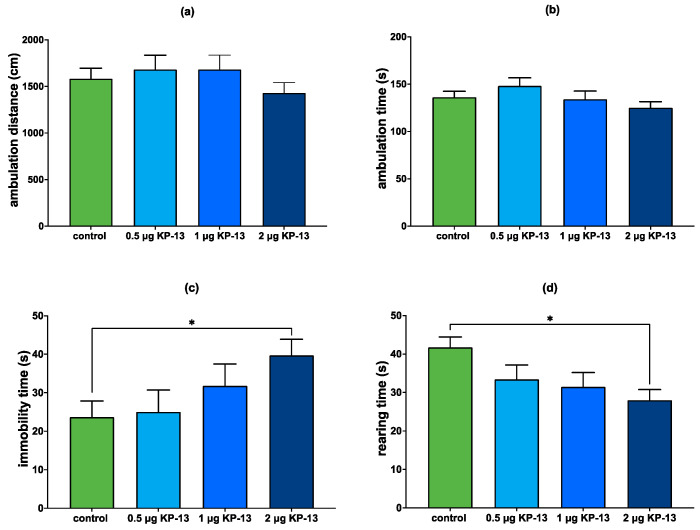
Ambulation in the open field test: (**a**) total distance traveled; (**b**) total time of ambulation; (**c**) immobility time; (**d**) number or rearings; mean + SEM, *n* = 4–6; * *p* < 0.05; Abbreviation: KP-13: kisspeptin-13.

**Figure 8 biomedicines-11-02446-f008:**
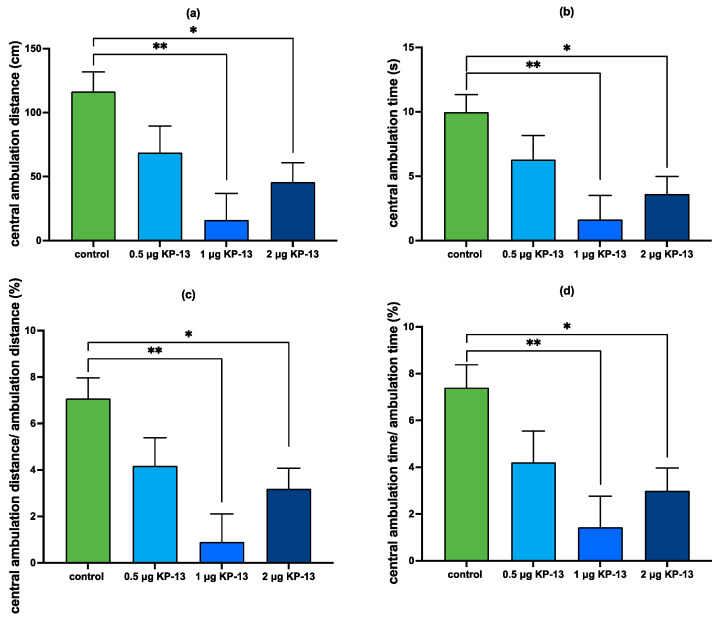
Central ambulation in the open field test: (**a**) distance traveled in the center of the arena; (**b**) time spent in the center of the arena; (**c**) central ambulation distance/total ambulation distance %; (**d**) central ambulation time/total ambulation time %; mean + SEM, *n* = 4–6; * *p* < 0.05, ** *p* < 0.01; Abbreviation: KP-13: kisspeptin-13.

**Figure 9 biomedicines-11-02446-f009:**
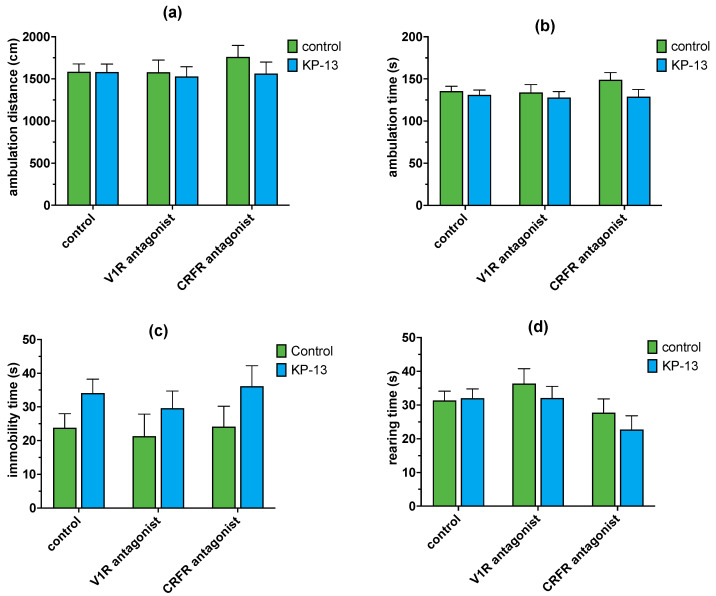
Ambulation in the open field test in the presence of V1R and CRFR antagonists in the open field test: (**a**) total distance traveled; (**b**) total time of ambulation; (**c**) immobility time; (**d**) number or rearings; mean + SEM, *n* = 6–15; Abbreviations: V1R: V1 receptor, CRFR: corticotropin-releasing factor receptor, KP-13: kisspeptin-13.

**Figure 10 biomedicines-11-02446-f010:**
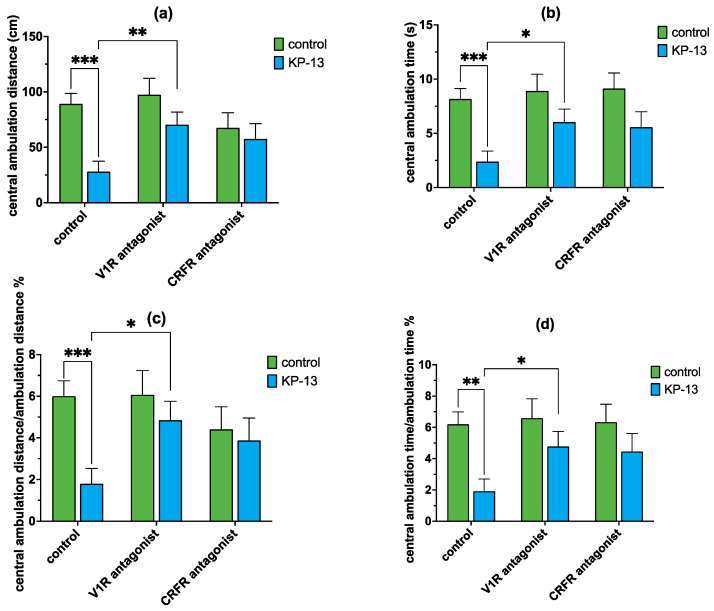
Central ambulation in the presence of V1R and CRFR antagonists in the open field test: (**a**) distance traveled in the center of the arena; (**b**) time spent in the center of the arena; (**c**) central ambulation distance/total ambulation distance %; (**d**) central ambulation time/total ambulation time %; mean + SEM, *n* = 6–15; * *p* < 0.05, ** *p* < 0.01, *** *p* < 0.001; Abbreviations: V1R: V1 receptor, CRFR: corticotropin-releasing factor receptor, KP-13: kisspeptin-13.

**Table 1 biomedicines-11-02446-t001:** Custom primers.

Genes	Forward (5′ → 3′)	Reverse (5′ → 3′)
*Avp*	CTG ACA TGG AGC TGA GAC AGT	CGC AGC TCT CGT CGC T
*Avpr1a*	TGG ACC GAT TCA GAA AAC CCT	GTT GGG CTC CGG TTG TTA GA
*Avpr1b*	CAG CAT AGG AGC CAA CCA TCA A	GAA AGC CCA GCT AAG CCG T
*Crf*	TGG TGT GGA GAA ACT CAG AGC	CAT GTT AGG GGC GCT CTC TTC
*Crfr1*	CGA AGA GAA GAA GAG CAA AGT ACA C	GCG TAG GAT GAA AGC CGA GA
*Crfr2*	CCC GAA GGT CCC TAC TCC TA	CTG CTT GTC ATC CAA AAT GGG T
*Gapdh*	CGG CCA AAT CTG AGG CAA GA	TTT TGT GAT GCG TGT GTA GCG

Abbreviations: Avp: Arginine vasopressin, Avpr1a: arginine vasopressin receptor 1A, Avpr1b: arginine vasopressin 1B, Crf: corticotropin-releasing factor, Crfr1: corticotropin-releasing factor receptor 1, Crfr2: corticotropin-releasing factor receptor 2, Gapdh: glyceraldehyde 3-phosphate dehydrogenase.

**Table 2 biomedicines-11-02446-t002:** Real-time polymerase chain reaction cycling protocol.

Steps	Temperature °C	Time	Number of Cycles
Uracil DNA glycosylase pretreatment	50	2 min	1
Initial denaturation	95	10 min	1
Denaturation	95	15 s	40
Annealing	60	30 s	
Extension	72	30 s	

## Data Availability

The data that support the findings of this study are available from the corresponding author K.C. and uploaded in the Mendeley repository.

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
