# Peer review of "A Brain Region-Dependent Alteration in the Expression of Vasopressin, Corticotropin-Releasing Factor, and Their Receptors Might Be in the Background of Kisspeptin-13-Induced Hypothalamic-Pituitary-Adrenal Axis Activation and Anxiety in Rats"

_biomedicines, 2023, doi:10.3390/biomedicines11092446_

Round 1
Reviewer 1 Report
This article is interesting and relevant. The authors carried out a lot of work using a large number of experimental animals, which was approved by the local ethics committee. However, the manuscript needs a serious technical revision.
Title: minimize the use of abbreviations.
Abstract: systematize the abstract, including the purpose, materials and methods, results, conclusion; all abbreviations should be written in full when first used.
Keywords: reduce the number of abbreviations; I recommend writing the terms in full to increase the citation of the article if it is accepted for publication.
Lines 26, 29, 33, 39, 58, 62, etc. - write all the abbreviations indicated in the text in full when using them for the first time. Avoid using abbreviations if they are used 4 or less times in the manuscript.
Please, state clearly the purpose of your study.
Materials and methods: please, add a flowchart explaining the design of your study.
Tables 1 and 2: add a note under the table explaining all the abbreviations used; I do not recommend using abbreviations in the names of tables here and further in the manuscript.
Line 221, 225-229, etc.: write the names of the genes in italics.
Figures 1, 2: add units of measurement for the vertical axes of all graphs.
Figure 3: I do not recommend using abbreviations in the name.
Add notes to all the figures where you explain all the abbreviations used.
Discussion: What is the clinical significance of the results obtained? What are the prospects for translating the results into clinical practice? What are the limitations of this study?
References: 32 out of 71 references, published more than 10 years ago. I recommend updating the analysis of previously conducted studies. What's new?
A minimal revision of the English language is required.
Author Response
First of all, we would like to thank the reviewer for the very meticulous review. We corrected the technical mistakes that were pointed out and tried to add the information that the reviewer asked for. We believe that it greatly increased the value of our manuscript.
See below our detailed answers to the questions/issues raised.
This article is interesting and relevant. The authors carried out a lot of work using a large number of experimental animals, which was approved by the local ethics committee. However, the manuscript needs a serious technical revision.
1. Title: minimize the use of abbreviations.
We thank the reviewer, and we corrected it accordingly.
2. Abstract: systematize the abstract, including the purpose, materials and methods, results, conclusion; all abbreviations should be written in full when first used.
We agree with the reviewer that it would be much easier for the reader if the abstract were systematized. However, due to the limit of the abstract word count, we could not add it.
We took out most of the abbreviations though, even with this we are over 250 word count, unfortunately. Hopefully, it is acceptable as such.
3. Keywords: reduce the number of abbreviations; I recommend writing the terms in full to increase the citation of the article if it is accepted for publication.
We thank the reviewer, corrected it. There are no abbreviations in the keywords.
4. Lines 26, 29, 33, 39, 58, 62, etc. - write all the abbreviations indicated in the text in full when using them for the first time. Avoid using abbreviations if they are used 4 or less times in the manuscript.
We have checked the manuscript for unjustified abbreviations (removed HPG axis, CNS) and wrote them in full when first used (HPA axis). See the manuscript for details, changes are indicated in red.
5. Please, state clearly the purpose of your study.
We are somewhat surprised that the reviewer found the purpose of the study not clear. Nevertheless, we have rephrased it to make it more clear:
“All the above-mentioned data highlight the important role these neuropeptides play in stress response and stress-related behavior. Based on this and our previous experiments [19,20], we hypothesized that KP-13 might alter the CRF and AVP signaling in the amygdala and hippocampus, two brain areas that are involved in the regulation of anxiety-like behavior and express KP and its receptors, as well as CRF and AVP and their receptors [34]. Therefore, the purpose of the present study was to assess if KP influences the CRF and AVP expression in the amygdala and hippocampus and if these two stress hormones might mediate KP’s anxiety- and HPA axis-inducing effects. First, the expressions of Crf, Crfr1, Crfr2, Avp, Avpr1a, and Avpr1b were measured after KP-13 treatment to assess if KP-13 influences the expression of these genes in the amygdala and hippocampus. Next, we also determined CRF and AVP protein contents in these brain regions. To see if they might mediate KP-13’s anxiety-inducing and HPA-activating effect, animals were pretreated with a non-selective CRF or VP antagonist, after which the behavior of the animals was recorded in a computerized open field test or trunk blood was collected to measure the plasma corticosterone level.”
6. Materials and methods: please, add a flowchart explaining the design of your study.
We thank the reviewer for suggesting a flowchart of the methods. We added figures to the manuscript (See in revised manuscript).
7. Tables 1 and 2: add a note under the table explaining all the abbreviations used; I do not recommend using abbreviations in the names of tables here and further in the manuscript.
We have removed the abbreviations from the tables.
8. Line 221, 225-229, etc.: write the names of the genes in italics.
We thank the reviewer for pointing out this mistake. In our original version, it was italic, but somehow in the final uploaded version, it was not. We corrected it, see the revised manuscript.
9. Figures 1, 2: add units of measurement for the vertical axes of all graphs.
We thank the reviewer for this comment, probably it wasn’t clearly stated in the manuscript. In the case of the PCR studies, the ratio of each mRNA relative to the housekeeping gene was calculated using the 2-ΔΔCT method. Thus, in the graph the relative gene expression/fold change was shown, therefore it has no units of the vertical axis.
The 2–∆∆Ct method, is a widely used formula in order to calculate the relative fold gene expression of samples when performing qPCR (Livak and Schmittgen, 2001).
In the manuscript to make it clearer we added:
“The ratio of each mRNA relative to the housekeeping gene was calculated using the 2-ΔΔCT method, and the relative gene expressions were determined for data presentation. “
Livak KJ, Schmittgen TD. Analysis of relative gene expression data using real-time quantitative PCR and the 2(-Delta Delta C(T)) Method. Methods. 2001 Dec;25(4):402-8. doi: 10.1006/meth.2001.1262. PMID: 11846609.
10. Figure 3: I do not recommend using abbreviations in the name.
We thank the reviewer. We corrected it accordingly.
11. Add notes to all the figures where you explain all the abbreviations used.
We thank the reviewer. We added them to the figures.
12. Discussion: What is the clinical significance of the results obtained? What are the prospects for translating the results into clinical practice? What are the limitations of this study?
Thanks to the reviewer for pointing out a drawback of the manuscript. We have added a paragraph about both to the Discussion of the manuscript.
“It is somewhat difficult to assess the clinical significance of our findings in rodents. Our results taken together with those of others do indicate that kisspeptins have an influence on the regulation of the HPA axis and stress-associated behaviors. Thus, it is also possible that changes in KP signaling might be involved in the development of HPA axis-related pathologies (e.g. major depression, post-traumatic stress disorder, and anxiety disorders) (Rana et al., 2022; Holsboer and Ising, 2021). Several clinical trials are already on the way that are based on the modulation of KP signaling, therefore it is crucial to establish the full range of KP’s biological action and must be taken into account that it might influence mood (Hu et al., 2022). Nevertheless, further extensive research is needed to fully establish the mechanism of KP’s action on the HPA axis and stress-related behaviors.
Our results must be interpreted in the context of several limitations of our study. First, in our study, only male Wistar rats were used to avoid the sex differences and the effect of the hormonal changes associated with the estrous cycle. Notably, HPA axis responsiveness shows marked sex differences. In fact, in rodents, females exhibit a more notable corticosterone response in the presence of any stressors owing to the circulating estradiol levels (Oyola et al., 2017). Therefore, animal models of both sexes would’ve yielded more translatable results and in the future KP’s effect on the HPA axis and stress-related behaviors need to be addressed in female rats, as well.
Furthermore, in the present study, we have investigated the effect of KP-13 on the gene expression of AVP and its receptors as well as CRF and its receptors, however, only AVP and CRF were investigated at the protein levels.
Protein levels vary on a high dynamic range, regulated by the rate of translation and degradation, the former being the determining factor (Schwanhüusser et al., 2014). The rate of translation is regulated by multiple mechanisms, only one of which is transcription. Other mechanisms include the activity of eukaryotic initiating factors (EIFs), structural features (e.g. internal ribosome-entry sequences, upstream open reading frames, and secondary or tertiary RNA structures), RNA-binding proteins, as well as microRNAs and small interfering RNAs that are involved in the regulation of translation (Gebauer and Hentze, 2004). Any of these mechanisms might be in play, and thus, our gene expression results on the receptors might not translate to the protein levels.
Also, it must be noted that other brain regions may also be involved in mediating KP-13’s effect on the HPA axis and anxiety-like behavior. The amygdala and hippocampus were chosen due to the distribution data of KP and KISS1R and the crucial role these regions play in the regulation of the neuroendocrine stress system (Higo et al., 2016; Dedic et al., 2017). KP and KISS1R are expressed in other brain regions as well, however at a much lower expression level (Brailoiu et al., 2005; Higo et al., 2016). Still, it is possible that it exerts its effect via another route. “
Rana, T.; Behl, T.; Sehgal, A.; Singh, S.; Sharma, N.; Abdeen, A.; Ibrahim, S.F.; Mani, V.; Iqbal, M.S.; Bhatia, S.; et al. Exploring the Role of Neuropeptides in Depression and Anxiety. Prog Neuropsychopharmacol Biol Psychiatry 2022, 114, 110478, doi:10.1016/j.pnpbp.2021.110478.
Holsboer, F.; Ising, M. Hypothalamic Stress Systems in Mood Disorders. Handb Clin Neurol 2021, 182, 33–48, doi:10.1016/B978-0-12-819973-2.00003-4.
Hu, K.-L.; Chen, Z.; Li, X.; Cai, E.; Yang, H.; Chen, Y.; Wang, C.; Ju, L.; Deng, W.; Mu, L. Advances in Clinical Applications of Kisspeptin-GnRH Pathway in Female Reproduction. Reprod Biol Endocrinol 2022, 20, 81, doi:10.1186/s12958-022-00953-y.
Oyola MG, Handa RJ. Hypothalamic-pituitary-adrenal and hypothalamic-pituitary-gonadal axes: sex differences in regulation of stress responsivity. Stress. 2017 Sep;20(5):476-494. doi: 10.1080/10253890.2017.1369523. Epub 2017 Aug 31. PMID: 28859530; PMCID: PMC5815295.
Schwanhüusser, B.; Busse, D.; Li, N.; Dittmar, G.; Schuchhardt, J.; Wolf, J.; Chen, W.; Selbach, M. Global Quantification of Mammalian Gene Expression Control. Nature 2011, 473, 337–342, doi:10.1038/nature10098.
Gebauer, F.; Hentze, M.W. Molecular Mechanisms of Translational Control. Nature Reviews Molecular Cell Biology 2004, 5, 827–835, doi:10.1038/nrm1488.
Higo, S.; Honda, S.; Iijima, N.; Ozawa, H. Mapping of Kisspeptin Receptor MRNA in the Whole Rat Brain and Its Co-Localisation with Oxytocin in the Paraventricular Nucleus. Journal of Neuroendocrinology 2016, 28, 1–8, doi:10.1111/jne.12356.
Dedic, N.; Chen, A.; Deussing, J.M. The CRF Family of Neuropeptides and Their Receptors - Mediators of the Central Stress Response. Current Molecular Pharmacology 2017, 11, 4–31, doi:10.2174/1874467210666170302104053.
Brailoiu, G.C.; Dun, S.L.; Ohsawa, M.; Yin, D.; Yang, J.; Jaw, K.C.; Brailoiu, E.; Dun, N.J. KiSS-1 Expression and Metastin-like Immunoreactivity in the Rat Brain. Journal of Comparative Neurology 2005, 481, 314–329, doi:10.1002/cne.20350.
13. References: 32 out of 71 references, published more than 10 years ago. I recommend updating the analysis of previously conducted studies. What's new?
Since the discovery of kisspeptins and the original studies about its distribution and biological role in reproductive function were indeed first published more than 10 years ago, I believe it is not surprising that there are some older references in the manuscript. In most of the cases, we included the original article as well as a newer article. Nevertheless, we looked the manuscript over and changed some references to newer ones, and added some more recently published references.

Reviewer 2 Report
This is a basic original research work where the investigators studied the mediation of KP-13’s stress-evoking actions. They used male Wistar rats after icv KP-13 treatment and measured relative gene expressions of the CRF (Crf, Crfr1, Crfr2) and AVP (Avp, Avpr1a, Avpr1b) in the amygdala and hippocampus. They also assessed CRF and AVP protein content. In another group of animals, the researchers administered CRF or V1 receptor antagonists before introducing the KP-13. In this challenge, they measured an open field test or plasma corticosterone levels. Findings from this study showed that in the amygdala, KP-13 induced an upregulation of Avp and Avpr1b expression, and downregulation of Crf. In the hippocampus, the mRNA level of Crf increased and the level of Avpr1a decreased. AVP protein content was also elevated in the amygdala. Behavioral tests showed that KP-13 could evoke anxiety-like behavior in the open field test. This behavior was blocked by administration of the V1 receptor blocker. The plasma corticosterone level was reduced when CRF and V1 receptor blockers were used. The authors, based on these findings proposed that KP-13 alters the AVP and CRF signaling and that might be a player in anxiety-like behavior in animals.
Please address the following points in the revised version:
Why only adult male Wistar rats were employed and no female animals were used? The ARRIVE guidelines emphasize on use of both sexes to provide evidence for sex-related or lack of sex-related responses. Please elaborate.
The authors have mentioned in the method section that “The doses of the antagonists were selected based on previous dose-response studies, in which they had no effect per se on the investigated parameters”. Please add the reference for these previous studies.
The authors have mentioned “To determine plasma corticosterone concentrations, trunk blood was collected in heparinized tubes. The plasma corticosterone concentration was measured by the fluorescence assay described by Zenker and Bernstein as modified by Purves and Sirett”. Can the author add the sensitivity of the test?
For the statistic, the authors have used both parametric and non-parametric tests. This means that they have determined the normal distribution of data and based on normal and non-normal distribution selected the appropriate tests. Please add which normality test was used and the result of the test.
The authors have not mentioned which statistical package, or program they have used, for example, R, SPSS, etc. Please add.
Please add more details as to what reasons the amygdala and hippocampus were chosen among all other regions of the rat brain.
Please add a schematic of the experimental setup with a timeline and measures. This overview will help for a better understanding of what was performed at which time point. Would be desirable with a schematic presentation of the open field and ambulation parameter as how it is determined.
Please add the limitations of this study.
Please add the translatability and value of translation of this study's findings. For example, how these findings can help with anything in the clinic, drug development, etc.
Author Response
Reviewer 2
First of all, we would like to thank the reviewer for the very meticulous review. We tried to add the information that the reviewer asked for. We believe that it greatly increased the value of our manuscript.
Find below our detailed answers to the questions/issues raised.
1. Why only adult male Wistar rats were employed and no female animals were used? The ARRIVE guidelines emphasize on use of both sexes to provide evidence for sex-related or lack of sex-related responses. Please elaborate.
Yes, in our study, only male Wistar rats were used to avoid the sex differences and the effect of the hormonal changes associated with the estrous cycle. Notably, HPA axis responsiveness shows marked sex differences (Oyola and Handa, 2017). In fact, in rodents, females exhibit a more notable corticosterone response in the presence of any stressors, owing to the circulating estradiol levels. Therefore, animal models of both sexes definitely need to be addressed. We do plan, actually, experiments are being conducted in our Lab now to see how KP-13 affects the HPA axis and stress-related behaviors. However, we definitely need to account for the estrous cycle, therefore more animals are needed.
Nevertheless, we believe that, though it somewhat reduces the value of the manuscript, the results conducted in male rats are still relevant. Still, this is one of the limitations of our study that we included in the Discussion (see revised manuscript).
Oyola MG, Handa RJ. Hypothalamic-pituitary-adrenal and hypothalamic-pituitary-gonadal axes: sex differences in regulation of stress responsivity. Stress. 2017 Sep;20(5):476-494. doi: 10.1080/10253890.2017.1369523. Epub 2017 Aug 31. PMID: 28859530; PMCID: PMC5815295.
2. The authors have mentioned in the method section that “The doses of the antagonists were selected based on previous dose-response studies, in which they had no effect per se on the investigated parameters”. Please add the reference for these previous studies.
In the literature, alpha-helical CRF(9-41) is used intracerebroventricularly in a range of doses (appr from 0.5 to 10 mcg), respectively. Based on some previously published articles (Kask et al., 1997; Heinrichs et al., 1994) pilot experiments were conducted at our Lab and found that a 1 mcg dose is sufficient: it alone did not affect the parameters studied at our lab, but were effective as an antagonist in other studies conducted with different neuropeptides. Since then several articles have been published by our Lab with alpha-helical CRF(9-41) (e.g. Telegdy and Adamik, 2008; Telegdy and Adamik, 2014; Jászberényi et al., 2004).
In the case of vasopressin receptor 1 antagonist, we relied on the literature data as well (Caltabiano et al., 1988; Drago et al., 1997), did a pilot study, and applied the smallest dose that per se did not affect the studied parameter and it did effectively block the effect of UCN I in our previous study (Bagosi et al., 2014).
As requested by the reviewer we added the references to the manuscript (see revised manuscript).
Telegdy G, Adamik A. Mediators involved in the hyperthermic action of neuromedin U in rats. Regul Pept. 2014 Jun-Aug;192-193:24-9. doi: 10.1016/j.regpep.2014.07.004. Epub 2014 Aug 7. PMID: 25108055.
Jászberényi M, Bujdosó E, Telegdy G. Behavioral, neuroendocrine and thermoregulatory actions of apelin-13. Neuroscience. 2004;129(3):811-6. doi: 10.1016/j.neuroscience.2004.08.007. PMID: 15541902.
Telegdy G, Tiricz H, Adamik A. Involvement of neurotransmitters in urocortin-induced passive avoidance learning in mice. Brain Res Bull. 2005 Oct 15;67(3):242-7. doi: 10.1016/j.brainresbull.2005.07.008. PMID: 16144661.
Bagosi Z, Csabafi K, Palotai M, Jászberényi M, Földesi I, Gardi J, Szabó G, Telegdy G. The effect of urocortin I on the hypothalamic ACTH secretagogues and its impact on the hypothalamic-pituitary-adrenal axis. Neuropeptides. 2014 Feb;48(1):15-20. doi: 10.1016/j.npep.2013.11.002. Epub 2013 Nov 21. PMID: 24331779.
The authors have mentioned “To determine plasma corticosterone concentrations, trunk blood was collected in heparinized tubes. The plasma corticosterone concentration was measured by the fluorescence assay described by Zenker and Bernstein as modified by Purves and Sirett”. Can the author add the sensitivity of the test?
We thank the reviewer for the question. We did not alter this method of corticosterone measurement from the two original articles.
The method described by Zenker and Bernstein utilizes the sulfuric acid-induced fluorescence of corticosterone. The original article investigated the specificity by examining the fluorescence of other steroids and also reported sensitivity values.
Firstly, at a ratio of 2.4 volume of sulphuric acid to 1.0 volume of 50% ethanol, apart from corticosterone, only hydrocortisone and estradiol fluoresce. The fluorescence of corticosterone is about five times as much as that of hydrocortisone, and adrenectomy abolishes fluorescence, which confirms that corticosterone is responsible for most of the fluorescent signal. The fluorescence of estradiol was found to be negligible in comparison.
Secondly, the following data were provided regarding sensitivity:
„the results obtained with 1.0 ml. of plasma were 33.3 ±1.2 γ per 100 ml. of corticosterone with a recovery of 88.8 per cent. These do not differ significantly from those obtained with 0.2 ml. of plasma, i.e. 34.5 ± 1.0 γ per 100 ml. with a 91.0 per cent recovery. Quantities as small as 0.2 ml. are clearly sufficient for analysis under these conditions. In order to test the reliability of the method, samples of a rat plasma pool (Long-Evans male rats) were placed in small test tubes and stored in the frozen state. Determinations made on 8 different days showed a mean of 29.9 γ of corticosterone per 100 ml. with a standard deviation of 1.15 and 95 per cent confidence limits of 2.6 γ per 100 ml.” (Zenker and Bernstein, 1958)
In the cited section, the Greek letter gamma (γ) is an obsolete non-SI unit that stands for μg (Butcher, 2023).
Purves and Sirett have measured corticosterone in response to corticotrophin (i.e. ACTH) in dexamethasone-treated rats (Purves and Sirett, 1965). Our group has followed their instructions for corticosterone measurement (their method is a slightly modified version of the original by Zenker and Bernstein). The article has reported an average index of precision (λ) of 0.185±0.010, calculated from 13 assays, which is considered sufficiently low and regular. The index of precision is an estimate of the standard deviation of the logarithm of the individual doses, calculated by dividing the standard deviation of the responses by the slope of the line connecting the response with the logarithm of the dose. Preferably the average of multiple assays is used (Gaddum, 1953).
Zenker, N. & Bernstein, D. E. The estimation of small amounts of corticosterone in rat plasma. J. Biol. Chem. 231, 695–701 (1958).
Butcher, T. Checking the Net Contents of Packaged Goods. NIST HB 133-2023 https://nvlpubs.nist.gov/nistpubs/hb/2023/NIST.HB.133-2023.pdf (2023) doi:10.6028/NIST.HB.133-2023.
Purves, H. D. & Sirett, N. E. Assay of corticotrophin in dexamethasone-treated rats. Endocrinology 77, 366–374 (1965).
Gaddum, J. H. Bioassays and Mathematics. Pharmacol Rev 5, 87–134 (1953).
For the statistic, the authors have used both parametric and non-parametric tests. This means that they have determined the normal distribution of data and based on normal and non-normal distribution selected the appropriate tests. Please add which normality test was used and the result of the test.
We made our decision if ANOVA was appropriate by assessing multiple parameters such as the histograms, skewness and kurtosis (excess) values, the normality test Kolmogorov-Smirnov and Levene’s test of homogeneity.
In the case of the PCR data, due to the control group (that is 1 since relative gene expression was calculated) none of the data passed normality (KS statistic: 1.000, Sig. 0.0001).
In the case of the ELISA data, we concluded a normal distribution (KS statistic: 0.173, Sig. 0.077).
In the case of the open-field tests: ambulation distance (KS statistic: 0.130, Sig. 0.086), ambulation time (KS statistic: 0.085, Sig. 0.20), immobility time (KS statistic: 0.105, Sig. 0.20), rearing (KS statistic: 0.146, Sig. 0.081), central ambulation distance (KS statistic: 0.164, Sig. 0.008), central ambulation time (KS statistic: 0.159, Sig. 0.052), central ambulation distance/total ambulation distance% (KS statistic: 0.134, Sig. 0.066) and central ambulation time/total ambulation time % (KS statistic: 0.144, Sig. 0.037; skewness and kurtosis between -1 and +1).
Based on these results we performed a nonparametric test for central ambulation distance. Central ambulation time/total ambulation time was significant, however, skewness and kurtosis were not as bad, therefore, we made the decision to analyze the data with a parametric test. Also, except for central ambulation distance (Levene statistic: 3.452, Sig. 0.026), all other data passed Levene’s test. ANOVA is robust enough to allow for failed normality with only a slight skewness and no far outliers (Blanca et al., 2016), however, unequal variance makes the data of central ambulation distance not suitable for ANOVA.
In the case of open-field tests with antagonists: ambulation distance (KS statistics: 0.051, Sig. 0.20), ambulation time (KS statistic: 0.93, Sig. 0.20), immobility time (KS statistic: 0.161, Sig. 0.001), rearing (KS statistic: 0.085, Sig. 0.20), central ambulation distance (KS statistic: 0.103, Sig. 0.186), central ambulation time (KS statistic: 0.089, Sig. 0.2), central ambulation distance/total ambulation distance% (KS statistic: 0.137, Sig. 0.070) and central ambulation time/total ambulation time % (KS statistic: 0.098, Sig. 0.2). In the case of immobility, KS test was significant, however, skewness and kurtosis were between -1 and +1, and the data passed the Levene’s test (Levene statistic: 2.011 Sig. 0.092), therefore, we performed ANOVA on the data.
In the case of corticosterone measurement data, we conducted ANOVA despite the failed normality (KS statistic: 0.129, Sig. 0.023) since skewness and kurtosis were between -1 and +1, and the data passed the Levene’s test (Levene statistic: 2.013 Sig. 0.093).
We added the following sentence to the Methods/Statistical analysis:
“The prerequisites of ANOVA were assessed via histograms, skewness and kurtosis, the Kolmogorov-Smirnov and Levene’s tests. “
And we added the following sentence to the Results section of the manuscript:
“The relative expression of Avp, Avpr1a, Avpr1b, Crf, Crfr1, and Crfr2 genes was calculated compared to Gapdh expression and analyzed by the Mann-Whitney test since the normality test and histograms showed a nonnormal distribution of the data.”
Blanca MJ, Alarcón R, Arnau J, Bono R, Bendayan R. Non-normal data: Is ANOVA still a valid option? Psicothema. 2017 Nov;29(4):552-557. doi: 10.7334/psicothema2016.383. PMID: 29048317.
The authors have not mentioned which statistical package, or program they have used, for example, R, SPSS, etc. Please add.
We conducted the statistical analysis with SPSS. As requested, we added the following sentence to the Methods/Statistics section of the manuscript:
“The statistical analysis was carried out by SPSS. “
Please add more details as to what reasons the amygdala and hippocampus were chosen among all other regions of the rat brain.
It is well established in the literature that the amygdala and hippocampus play a central role in the adaptation to stress.
Distribution data of the kisspeptin system revealed a relatively high expression in the amygdala and hippocampus.
In our previous study, kisspeptin evoked an elevation of corticosterone and anxiety-like behavior.
Putting these data together, it was plausible that kisspeptin might act in these two regions to mediate its actions on the HPA axis and stress-related behavior.
Kisspeptin is expressed in other brain regions as well, however at a much lower expression level. Still, it is possible that it exerts its effect via another route. This is one of the limitations of the study that we included in the manuscript as well.
To make the connection and our choice clear we added the following paragraph to the Introduction:
“CRF and AVP are expressed in abundance in the amygdala and hippocampus. In fact, the highest expression of CRF outside of the hypothalamus is found in the amygdala (Callahan et al., 2013). Furthermore, CRF is coexpressed in subpopulations of hippocampal interneurons throughout the hippocampal layers. These CRF-neurones activate upon stress and mediate stress-induced effects of the hippocampus (Gunn et al., 2019; Chen et al., 2004). Hypothalamic AVP-expressing fibers project to the amygdala, which expresses both V1aR and V1bR to exert its stress-inducing effect (Neugebauer et al., 2020; Tong et al., 2021). Also, AVP-producing neurons are found in the amygdala (Neugebauer et al., 2020). AVP signaling is also involved in the regulation of hippocampal processes and consequently stress-related behaviors (Cilz et al., 2019; Zagraen et al., 2022).”
And we added the following sentences to the Discussion:
“Also, it must be noted that other brain regions may also be involved in mediating KP-13’s effect on the HPA axis and anxiety-like behavior. The amygdala and hippocampus were chosen due to the distribution data of KP and KISS1R and the crucial role these regions play in the regulation of the neuroendocrine stress system (Higo et al., 2016; Dedic et al., 2017). KP and KISS1R are expressed in other brain regions as well, however at a much lower expression level (Brailoiu et al., 2005; Higo et al., 2016). Still, it is possible that it exerts its effect via another route. “
Callahan, L.B.; Tschetter, K.E.; Ronan, P.J. Inhibition of Corticotropin Releasing Factor Expression in the Central Nucleus of the Amygdala Attenuates Stress-Induced Behavioral and Endocrine Responses. Front Neurosci 2013, 7, 195, doi:10.3389/fnins.2013.00195.
Paretkar, T.; Dimitrov, E. The Central Amygdala Corticotropin-Releasing Hormone (CRH) Neurons Modulation of Anxiety-like Behavior and Hippocampus-Dependent Memory in Mice. Neuroscience 2018, 390, 187–197, doi:10.1016/j.neuroscience.2018.08.019.
Gunn, B.G.; Sanchez, G.A.; Lynch, G.; Baram, T.Z.; Chen, Y. Hyper-Diversity of CRH Interneurons in Mouse Hippocampus. Brain Struct Funct 2019, 224, 583–598, doi:10.1007/s00429-018-1793-z.
Chen, Y.; Brunson, K.L.; Adelmann, G.; Bender, R.A.; Frotscher, M.; Baram, T.Z. Hippocampal Corticotropin Releasing Hormone: Pre- and Postsynaptic Location and Release by Stress. Neuroscience 2004, 126, 533–540, doi:10.1016/j.neuroscience.2004.03.036.
Neugebauer, V.; Mazzitelli, M.; Cragg, B.; Ji, G.; Navratilova, E.; Porreca, F. Amygdala, Neuropeptides, and Chronic Pain-Related Affective Behaviors. Neuropharmacology 2020, 170, 108052, doi:10.1016/j.neuropharm.2020.108052.
Tong, W.H.; Abdulai-Saiku, S.; Vyas, A. Arginine Vasopressin in the Medial Amygdala Causes Greater Post-Stress Recruitment of Hypothalamic Vasopressin Neurons. Molecular Brain 2021, 14, 141, doi:10.1186/s13041-021-00850-2.
Cilz, N.I.; Cymerblit-Sabba, A.; Young, W.S. Oxytocin and Vasopressin in the Rodent Hippocampus. Genes, Brain and Behavior 2019, 18, 1–14, doi:10.1111/gbb.12535.
Zagrean, A.-M.; Georgescu, I.-A.; Iesanu, M.I.; Ionescu, R.-B.; Haret, R.M.; Panaitescu, A.M.; Zagrean, L. Oxytocin and Vasopressin in the Hippocampus. Vitam Horm 2022, 118, 83–127, doi:10.1016/bs.vh.2021.11.002.
Higo, S.; Honda, S.; Iijima, N.; Ozawa, H. Mapping of Kisspeptin Receptor MRNA in the Whole Rat Brain and Its Co-Localisation with Oxytocin in the Paraventricular Nucleus. Journal of Neuroendocrinology 2016, 28, 1–8, doi:10.1111/jne.12356.
Dedic, N.; Chen, A.; Deussing, J.M. The CRF Family of Neuropeptides and Their Receptors - Mediators of the Central Stress Response. Current Molecular Pharmacology 2017, 11, 4–31, doi:10.2174/1874467210666170302104053.
Brailoiu, G.C.; Dun, S.L.; Ohsawa, M.; Yin, D.; Yang, J.; Jaw, K.C.; Brailoiu, E.; Dun, N.J. KiSS-1 Expression and Metastin-like Immunoreactivity in the Rat Brain. Journal of Comparative Neurology 2005, 481, 314–329, doi:10.1002/cne.20350.
Please add a schematic of the experimental setup with a timeline and measures. This overview will help for a better understanding of what was performed at which time point. Would be desirable with a schematic presentation of the open field and ambulation parameter as how it is determined.
We thank the reviewer. We added two figures to the manuscript showing the experimental setup and the open-field apparatus with the parameters measured during the open-field test (see revised manuscript).
Please add the limitations of this study.
We thank the reviewer for the comment, we added the following to the manuscript:
“Our results must be interpreted in the context of several limitations of our study. First, in our study, only male Wistar rats were used to avoid the sex differences and the effect of the hormonal changes associated with the estrous cycle. Notably, HPA axis responsiveness shows marked sex differences. In fact, in rodents, females exhibit a more notable corticosterone response in the presence of any stressors owing to the circulating estradiol levels. Therefore, animal models of both sexes would’ve yielded more translatable results and in the future KP’s effect on the HPA axis and stress-related behaviors need to be addressed as well.
Furthermore, in the present study, we have investigated the effect of KP-13 on the gene expression of AVP and its receptors as well as CRF and its receptors, however, only AVP and CRF were investigated at the protein levels.
Protein levels vary on a high dynamic range, regulated by the rate of translation and degradation, the former being the determining factor (Schwanhüusser et al., 2014). The rate of translation is regulated by multiple mechanisms, only one of which is transcription. Other mechanisms include the activity of eukaryotic initiating factors (EIFs), structural features (e.g. internal ribosome-entry sequences, upstream open reading frames, and secondary or tertiary RNA structures), RNA-binding proteins, as well as microRNAs and small interfering RNAs that are involved in the regulation of translation (Gebauer and Hentze, 2004). Any of these mechanisms might be in play, and thus, our gene expression results on the receptors might not translate to the protein levels.
Also, it must be noted that other brain regions may also be involved in mediating KP-13’s effect on the HPA axis and anxiety-like behavior. The amygdala and hippocampus were chosen due to the distribution data of KP and KISS1R and the crucial role these regions play in the regulation of the neuroendocrine stress system (Higo et al., 2016; Dedic et al., 2017). KP and KISS1R are expressed in other brain regions as well, however at a much lower expression level (Brailoiu et al., 2005; Higo et al., 2016). Still, it is possible that it exerts its effect via another route. “
Schwanhüusser, B.; Busse, D.; Li, N.; Dittmar, G.; Schuchhardt, J.; Wolf, J.; Chen, W.; Selbach, M. Global Quantification of Mammalian Gene Expression Control. Nature 2011, 473, 337–342, doi:10.1038/nature10098.
Gebauer, F.; Hentze, M.W. Molecular Mechanisms of Translational Control. Nature Reviews Molecular Cell Biology 2004, 5, 827–835, doi:10.1038/nrm1488.
Higo, S.; Honda, S.; Iijima, N.; Ozawa, H. Mapping of Kisspeptin Receptor MRNA in the Whole Rat Brain and Its Co-Localisation with Oxytocin in the Paraventricular Nucleus. Journal of Neuroendocrinology 2016, 28, 1–8, doi:10.1111/jne.12356.
Dedic, N.; Chen, A.; Deussing, J.M. The CRF Family of Neuropeptides and Their Receptors - Mediators of the Central Stress Response. Current Molecular Pharmacology 2017, 11, 4–31, doi:10.2174/1874467210666170302104053.
Brailoiu, G.C.; Dun, S.L.; Ohsawa, M.; Yin, D.; Yang, J.; Jaw, K.C.; Brailoiu, E.; Dun, N.J. KiSS-1 Expression and Metastin-like Immunoreactivity in the Rat Brain. Journal of Comparative Neurology 2005, 481, 314–329, doi:10.1002/cne.20350.
Please add the translatability and value of translation of this study's findings. For example, how these findings can help with anything in the clinic, drug development, etc.
We have added a paragraph to the Discussion of the manuscript.
“It is somewhat difficult to assess the clinical significance of our findings in rodents. Our results taken together with those of others do indicate that kisspeptins have an influence on the regulation of the HPA axis and stress-associated behaviors. Thus, it is also possible that changes in KP signaling might be involved in the development of HPA axis-related pathologies (e.g. major depression, post-traumatic stress disorder, and anxiety disorders) (Rana et al., 2022; Holsboer and Ising, 2021). Several clinical trials are already on the way that are based on the modulation of KP signaling, therefore it is crucial to establish the full range of KP’s biological action and must be taken into account that it might influence mood (Hu et al., 2022). Nevertheless, further extensive research is needed to fully establish the mechanism of KP’s action on the HPA axis and stress-related behaviors."
Rana, T.; Behl, T.; Sehgal, A.; Singh, S.; Sharma, N.; Abdeen, A.; Ibrahim, S.F.; Mani, V.; Iqbal, M.S.; Bhatia, S.; et al. Exploring the Role of Neuropeptides in Depression and Anxiety. Prog Neuropsychopharmacol Biol Psychiatry 2022, 114, 110478, doi:10.1016/j.pnpbp.2021.110478.
Holsboer, F.; Ising, M. Hypothalamic Stress Systems in Mood Disorders. Handb Clin Neurol 2021, 182, 33–48, doi:10.1016/B978-0-12-819973-2.00003-4.
Hu, K.-L.; Chen, Z.; Li, X.; Cai, E.; Yang, H.; Chen, Y.; Wang, C.; Ju, L.; Deng, W.; Mu, L. Advances in Clinical Applications of Kisspeptin-GnRH Pathway in Female Reproduction. Reprod Biol Endocrinol 2022, 20, 81, doi:10.1186/s12958-022-00953-y.

Round 2
Reviewer 1 Report
The authors have modified the manuscript, but there are several minor comments.
Please add a Note to Figure 1 and explain all the abbreviations you used.
Line 380 - Delete the extra colon after the name of Figure 9.
Author Response
Dear Reviewer 1,
Thank you for the 2nd review. We have added the abbreviations to Figure 1 and erased the extra colon in Figure 9. See the revised manuscript uploaded.
I hope you find the manuscript in the present form acceptable.
Regards,
Reviewer 2 Report
The authors have taken into account the points raised by this reviewer and have addressed the points in the revised version. Thanks. There is no further comment-suggestion.
Author Response
Dear Reviewer 2,
Again thank you for your review. We appreciated the raised questions and issues and are thankful that you found our answers adequate.
We hope that you find the manuscript in the present form acceptable for publication.
Best regards,